# Assessing Phytogenic and Chemogenic Silver Nanoparticles for Antibacterial Activity and Expedited Wound Recuperation

**DOI:** 10.3390/nano14030237

**Published:** 2024-01-23

**Authors:** Bilal Ahmad, Li Chang, Caiyun Yin, Zhou Wu, Aidi Tong, Chunyi Tong, Bin Liu

**Affiliations:** 1College of Biology, Hunan University, Changsha 410082, China; bilalahmad_271@live.com (B.A.); changli@caas.cn (L.C.); yincaiyun@hnu.edu.cn (C.Y.); wuzhou@hnu.edu.cn (Z.W.); 2College of Chemistry and Chemical Engineering, Hunan University, Changsha 410082, China; 3Institute of Bast Fiber Crop, Chinese Academy of Agriculture Sciences, Changsha 410082, China; 4School of Medicine, Hunan Normal University, Changsha 410082, China; 202120193331@hunnu.edu.cn

**Keywords:** silver nanoparticles, green synthesis, antibacterial, antioxidant, coriander

## Abstract

Green silver nanoparticles (AgNPs) possess tremendous promise for diverse applications due to their versatile characteristics. Coriander and other plant extracts have become popular for greenly synthesizing AgNPs as an economical, biocompatible, cost-effective, and environmentally beneficial alternative to chemical processes. In this study, we synthesized AgNPs from coriander leaves and evaluated their antibacterial, anti-inflammatory, antioxidant, and wound-healing acceleration properties in comparison to chemically synthesized AgNPs. The zeta potentials of AgNPs extracted from green and chemical processes were −32.4 mV and −23.4 mV, respectively. TEM images showed a cuboidal shape of green and chemical AgNPs with a diameter of approximately 100 nm. The FTIR spectra of green AgNPs showed an extreme absorption peak at 3401 cm^−1^, which signifies O-H stretching vibrations, typically linked to hydroxyl groups. In vitro results elaborated that AgNPs from coriander exerted a stronger effect on anti-*Klebsiella pneumoniae* (*KP*) through interrupting cell integrity, generating ROS, depleting ATP, and exhibiting significant antioxidant activity, compared with AgNPs synthesized chemically. In vivo experiments showed that AgNPs from coriander, as opposed to chemically manufactured AgNPs, greatly accelerated the healing of wounds contaminated with *Klebsiella pneumoniae* bacteria by effectively eliminating the bacteria on the wounds and stimulating skin regeneration and the deposition of dense collagen. In vivo assays further demonstrated that green AgNPs effectively enhanced *Klebsiella pneumoniae*-infected wound healing by extenuating local inflammatory responses and up-regulating VEGF and CD31 expression. In conclusion, green AgNPs significantly alleviated the inflammation without significantly harming the organism.

## 1. Introduction

Nanotechnology is a captivating multidisciplinary domain within the realm of science and technology undergoing rapid advancement [1]. Nanoparticles (NPs) of gracious metals like platinum, gold, and silver are well known to have large applications in electronics, magnetics, optical receptors, and catalysts in chemical reaction [2]. AgNPs rank among the most renowned and extensively utilized metallic nanomaterials. They have found application in a wide range of fields, and their usage is on a continuous upward trajectory. In addition to their cost-effectiveness and straightforward production, the utilization of AgNPs is bolstered by the perception that silver is relatively safe due to its low toxicity and limited bioavailability [3].

There are many methods available for the synthesis of AgNPs, such as chemical [4], electrochemical [5], UV radiation [6], ultrasonic fields and photochemical methods [7,8], and aerosol technologies and biological techniques [9]. Many of the described techniques involve multiple additional steps, substantial energy consumption, low conversion of materials, purification challenges, and the use of hazardous chemicals [10,11]. Although chemical synthesis is a widely recognized method for generating AgNPs, it necessitates the use of harmful chemicals as both reducing and capping agents [12]. Meanwhile, the chemical synthesis of NPs can result in the presence of certain toxic chemical species absorbed onto them, potentially causing adverse effects in their applications [13]. Moreover, with the increasing use of chemical AgNPs, human exposure to them has risen. Research has indicated that AgNPs can enter the bloodstream and accumulate in various organs, potentially leading to liver or kidney toxicity when administered orally, through inhalation, or subcutaneously [14,15].

Given the drawbacks of the chemical synthesis approach, there is a necessity for the development of cost-effective, biocompatible, and environmentally friendly methods for producing green AgNPs to fulfill the increasing demand of biomedicine applications. Green and biological methods employing economical, efficient, and eco-conscious catalysts are gaining significant prominence [16]. Green, environmentally friendly nanoparticles are produced by the biosynthesis of nanoparticles using plant extracts, which also makes use of abundant plant resources that are widely distributed, easily accessible, safe to handle, and have a variety of metabolites [17].

AgNPs possess diverse applications, including antibacterial properties, water sterilization, and wound healing treatment [11]. Valodkar et al. have synthesized silver and copper nanoparticles from the latex of *Euphorbiaceae*. These nanoparticles have demonstrated remarkable antibacterial properties against both Gram-negative and Gram-positive bacteria [18]. Ismail Ocsoy et al. have described the synthesis of colloidal silver nanoparticles using a hydrothermal method using red cabbage extract. The synthesized particles exhibit a spherical shape and are characterized by a high yield and monodispersity of materials achieved using this method. Furthermore, the AgNPs acquired exhibit heightened antibacterial efficacy against *Staphylococcus aureus* (Gram-positive) and *Escherichia coli* (Gram-negative) pathogens [19]. Ghosh et al. have synthesized silver nanoparticles using the tuber extract of *Dioscorea bulbifera*. The synthesized materials exhibit diverse forms and demonstrate strong antibacterial action against test microorganisms, including *Pseudomonas aeruginosa*, *Escherichia coli*, and *Acinetobacter baumannii* [20]. AgNPs produced using the current technique exhibited potent antibacterial action due to their functionalization with the plant components of *Coriandrum sativum*. Philip et al. have presented a simple and fast environmentally friendly method for synthesizing spherical silver and gold nanoparticles. This synthesis method was carried out at room temperature and at a higher temperature of 373 K. The reducing and stabilizing agent used in this process was an extract from *Murraya koenigii* leaves [21]. Mittal et al. and Akhtar et al. have conducted a review on the synthesis of metallic nanoparticles (Ag, Au, Cu, Pt, Fe, Pd) utilizing plant extracts. These synthesized materials have shown bioactivity and are being explored for therapeutic uses [22,23]. Based on the previous findings, AgNPs produced from different plant extracts exhibited remarkable antibacterial properties against both Gram-positive and Gram-negative bacteria.

Currently, nanobiotechnology stands out as a captivating field of study within contemporary material science. It explores the structural aspects involving the use of plant extracts and various plant markers in the creation of nanoparticles. When these plant extracts are employed to produce AgNPs through the reduction and stabilization process, these nanoparticles lack chemical compounds on their surface, making them non-hazardous to human cells [8,9,10,11,12,13,14,15,16,17,18,19,20,21,22,23,24].

Coriander, scientifically known as *Coriandrum sativum* L., is a fragrant herb that belongs to the *Apiaceae* family. It boasts substantial nutritional and medicinal attributes. Primarily grown for its herbaceous qualities, the fresh leaves and ground seeds of this plant offer remarkable sensory characteristics. Coriander extracts and essential oils have demonstrated various health benefits, including antibacterial, antioxidant, free radical-fighting, antidiabetic, anti-inflammatory, anticonvulsant, anxiolytic, hypoglycemic, hypolipidemic, anticancer, and antimutagenic properties [25,26,27]. AgNPs extracted from coriander leaves have strong antimicrobial properties against many fungi, viruses, and bacteria due to their activity as photocatalysts and their ability to produce reactive oxygen species (ROS) [28].

In this study, we have developed a robust nanoplatform for synthesizing AgNPs with strong antibacterial and anti-inflammatory functions in Coriandrum sativum (green AgNPs). These nanoparticles were effective in combating *Klebsiella pneumoniae* (Gram-negative). We compared the antimicrobial efficacy of chemically synthesized silver nanoparticles (chemical AgNPs) with those synthesized using coriander leaf extract. Furthermore, we demonstrated the ability of this nanoplatform to eliminate multidrug-resistant bacteria in vitro, thanks to the inherent bactericidal properties of AgNPs derived from coriander leaf extract. We also conducted a comparative analysis of the characteristics and antioxidant activities of both green and chemically synthesized AgNPs. Moreover, we successfully applied this potent antibacterial nanoplatform for treating wounds infected with *Klebsiella pneumoniae*.

## 2. Materials and Methods

### 2.1. Materials and Reagents

Silver nitrate (AgNO_3_) was purchased from Sigma-Aldrich (St. Louis, MO, USA) and used as a precursor for the synthesis of AgNPs. Citric acid monohydrate and ascorbic acid were purchased from Sinopharm Chemical Reagent Co., Ltd. Shanghai, China. Coriander leaves were obtained from the biological garden of Hunan University, China. The Enhanced ATP Assay Kit (BHI, Shanghai, China) was utilized for the experiments. Millipore deionized water with a resistivity of 18.2 MΩ cm was used throughout all procedures. All other chemicals and solvents were of analytical grade.

### 2.2. Preparation of AgNPs by Green Method

AgNPs were prepared following our previous method [28]. Briefly, fresh coriander leaves (20 g) were obtained and washed thoroughly. Then, the leaves were dried at room temperature and crushed in 100 mL of double-distilled water (ddH_2_O). An oil bath was used for boiling selected leaves for 2 h at 100 °C. Then, the extract was filtered through 25-µm filter paper until a clear solution was obtained. AgNO_3_ (0.1 mM) solution was prepared in 100 mL of ddH_2_O. The solution was stirred at room temperature for 30 min. Then, 10 mL of plant extract was added dropwise into the AgNO_3_ solution. The stirring continued for 48 h in the dark. The color of the solution was changed to black. Then, the solution was centrifuged twice at 14,000 rpm for 15 min. The supernatant was discarded, and the samples were kept in a vacuum freeze dryer for 24 h for drying.

### 2.3. Preparation of AgNPs by Chemical Method

AgNPs were prepared by adding 1 mL of trisodium citrate (1%) and 0.3 mL of AgNO_3_ to a 5 mL amber glass bottle. Then, 1.2 mL of ddH_2_O was added and stirred for 5 min in the dark (solution A). Thereafter, ddH_2_O (47.5 mL) was added to a 100 mL conical flask and allowed to heat in an oil–water bath at 120 °C. Then, 50 µL of ascorbic acid was added and boiled for 2 min. Then, solution A was added to a conical flask and stirred for 1 h. The yellow color indicated the formation of AgNPs.

### 2.4. Characterizations

A UV-vis spectrophotometer (Beckman Coulter Inc., Brea, CA, USA) was used for recording UV-vis absorbance spectra of samples. TEM images were captured by a JEM-2100 microscope (JEOL, Tokyo, Japan, 200 kV). The size and zeta potential values of green and chemical AgNPs were measured by a Zetasizer Nano ZS (Malvern Ltd., Malvern, UK) in deionized water at 0.10 mg/mL of concentration. Thermo Fisher Scientific K-Alpha, USA was used for XPS analysis. Bruker Tensor 27 SpectraLab Scientific Inc., 38 McPherson St. Markham, ON, Canada was used for FTIR analysis. The HPLC analyses of the leaf extracts (coriander sativum) and the AgNPs from ascorbic acid and coriander were conducted on a Waters/ACQUTIY High Performance Liquid Chromatograph (Waters Corporation, Milford, CT, USA), PDA Detector (Waters Corporation), and Option R7 Ultra AND Ultrapure Water System (ELGA LabWaters, Lane End Business Park, Lane End, High Wycombe, UK).

### 2.5. In Vitro Antibacterial Experiment

*Klebsiella pneumoniae* was cultured for 24 h at 37 °C in Luria–Bertani broth (LB broth). The antibacterial impact of the nanomaterials was evaluated using the agar diffusion method. In summary, a 100 μL bacterial suspension containing 10^8^ CFU mL^−1^ of *Klebsiella pneumoniae* was evenly spread on the agar plate. Subsequently, 20 μL of the test samples (green AgNPs, chemical AgNPs, and water) were applied to a filter paper placed on the agar surface with *Klebsiella pneumoniae*. The agar plates were then placed in a 37 °C incubator. After 12 h of incubation, the sizes of the inhibition zones were measured to gauge the antibacterial effectiveness.

Minimum inhibitory concentration (MIC) of green and chemical AgNPs were performed on different concentrations. MICs of green and chemical AgNPs were identified using the micro-dilution method. Bacterial suspensions *Klebsiella pneumoniae* (1 × 10^4^ CFU mL^−1^) in 96-well plates were incubated with a series of concentrations (0, 2, 4, 6, 8, and 10µg/mL) of green and chemical AgNPs at 37 °C for 24 h. After incubation, the lowest concentration producing no visible bacterial growth was determined to be the MIC.

In vitro antibacterial activity of green and chemical AgNPs was performed at different concentrations with vancomycin as a control. The OD_600_ values were described and calculated at different concentrations. The plate counting method was employed to assess the in vitro antibacterial efficacy of the test samples, including water, green AgNPs, and chemical AgNPs. Subsequently, 500 μL of bacterial suspension with a concentration of 1 × 10^7^ CFU mL^−1^ was cultured with water as a control, green AgNPs, chemical AgNPs, and vancomycin. Following this, 20 μL of the appropriately diluted bacterial suspension was spread onto an LB agar plate, then incubated at 37 °C for 12 h. The count of bacterial-colony-forming units was conducted, and the antibacterial effectiveness was determined using the formula below.
Bacterial Survival ratio (%) = CFU_(Sample)_/CFU_(Control)_ × 100

### 2.6. Live/Dead Staining Assay

This study assessed bacterial viability using the Live/Dead Bacterial Viability Kit. *Klebsiella pneumoniae*, in various test groups (control, green AgNPs, chemical AgNPs, cefixime), was incubated at 37 °C for 2 h. Following incubation, the samples were subjected to centrifugation at 8000 rpm for 5 min at 40 °C and subsequently stained with Calcein-AM in the absence of light for 30 min. The bacteria were then thoroughly rinsed with PBS to eliminate any surplus dye. Finally, the bacterial solutions were applied to glass slides and examined using a Laser Scanning Confocal Microscope (FV1200, Olympus, Center Valley, PA, USA).

### 2.7. Reactive Oxygen Species (ROS) Detection Assay

The ROS level of *Klebsiella pneumoniae* cells was measured using a Reactive Oxygen Species Assay Kit. Briefly, *Klebsiella pneumoniae* cells (10^8^ CFU/mL) were incubated at 37 °C for 30 min. Then, *Klebsiella pneumoniae* cells were washed with PBS and incubated with a DCFH-DA probe for 30 min in the dark. Finally, all samples were imaged under Laser Scanning Confocal Microscope (FV1200, Olympus).

### 2.8. ATP Assay

The level of metabolic activity in bacteria was assessed using an Enhanced ATP Assay Kit from Beyotime (Haimen, China). Cells of *Klebsiella pneumoniae*, collected from various samples, were gathered by centrifugation at 8000 rpm for 5 min at 4 °C, then suspended in 100 μL of lysis buffer. Subsequently, the resulting supernatant was examined on a multimode detection platform in luminance mode.

### 2.9. Protein and Nucleic Acid Leakage Study

Our previous method was adopted to detect protein and nucleic acid leakage from *Klebsiella pneumoniae* induced by different samples [29]. Precisely, 10^9^ CFU mL^−1^
*Klebsiella pneumoniae* was firstly incubated with PBS (control), green AgNPs, chemical AgNPs, and cefixime for 3 h at 37 °C. Thereafter, the bacterial suspension of *Klebsiella pneumoniae* was centrifuged at 8000 rpm for 5 min at 4 °C, and the harvested supernatant was transferred to 1.5 mL Eppendorf tubes for NanoDrop (Waltham, MA, USA) 2000 Spectrophotometer reading.

### 2.10. In Vitro Biofilm Abolition Assay

To establish biofilms, *Klebsiella pneumoniae* cells at a concentration of 10^9^ colony-forming units per milliliter (CFU mL^−1^) were introduced into each well of a 24-well plate, with 1000 μL of the cell suspension in each well. The plates were then placed under stationary conditions at 37 °C for a period of 2 days to allow for biofilm formation. Once the biofilms had developed, the culture medium was removed, and the plates were carefully rinsed with phosphate-buffered saline (PBS) to eliminate any planktonic (free-floating) bacteria.

Subsequently, varying concentrations of green AgNPs (40 and 50 μg/mL) and chemical AgNPs (40 and 50 μg/mL) were introduced into the respective wells. In the control group, 200 μL of PBS, devoid of any nanomaterials, was employed. All the groups were then incubated at 37 °C for an additional 2 days. The media in each well was then removed, and the plates were washed carefully with PBS three times. The biofilm in each well was fixed with 500 μL methanol for 10 min and stained with 0.1% crystal violet for 30 min. After washing with PBS three times, 95% ethanol (1000 μL per well) was added to solubilize the crystal violet staining. Then, we measured OD_560_ in each well to determine the biofilm formation.

### 2.11. Enzyme-Based Antioxidant and DPPH Scavenging Assay

Assessment of enzyme-based activity on TMB was conducted as follows: In a 1 mL Eppendorf tube, 285 μL ddH_2_O, 2 μL TMB, the respective quantities of green or chemical AgNPs, and 3 μL of H_2_O_2_ were added and mixed with gentle shaking. The solution was incubated for 5 min before UV-vis spectrophotometer analysis.

The antioxidant activity of the synthesized AgNPs was measured by the free radical scavenging activity of the DPPH (1,1-diphenyl 2-picrylhydrazyl) method. A DPPH solution of 0.1 mM in DMSO was formulated. Subsequently, 1 mL of this DPPH solution was combined with 5 mL of various concentrations (5, 10, 15, 20, and 25 μg/mL) of both green and chemically synthesized AgNPs, which were prepared in water. This mixture underwent thorough agitation and was left to settle at room temperature for approximately 30 min. Following this incubation period, the absorbance at 517 nm was recorded using a UV-vis spectrophotometry.

### 2.12. In Vitro Cytotoxicity Assay

To investigate the in vitro cytotoxicity of green and chemical AgNPs, HUV-EC cells (1 × 10^4^) were seeded in a 96-well plate. After incubation for 24 h, the culture media was removed and fresh culture media comprising different concentrations of green and chemical AgNPs (0, 10, 20, 30, 40, 50, and 60 μg/mL). After 24 h incubation, the samples were washed with PBS three times and 100 μL of MTT solution was added into each well. After incubation for 2 h, cell viability was measured by a microplate reader in the absorbance of 490 nm.

Fresh BALB/c mouse blood was processed by centrifugation at 3500 rpm and 4 °C for 5 min to obtain fresh mouse erythrocytes. Various concentrations (2, 4, 6, 8, and 10 µg/mL) of green and chemical AgNPs were then mixed with 20 µL of erythrocytes to maintain a total system volume of 500 µL. After a 3 h incubation at 37 °C, the samples underwent the same centrifugation process, and photographs were taken to assess hemolysis. Following centrifugation, 200 µL of the supernatant was gently collected and its absorbance at 540 nm was measured using an enzyme-labelled detector for each sample. Negative control was achieved using PBS, while positive control used deionized water. The hemolysis rate was determined using the following equation:Hemolysis rate (%) = (A _Sample_ − A _Negative Control_)/(A _Positive Control_ − A _Negative Control_) × 100%

### 2.13. In Vivo Treatment of Wound Infection in BALB/c Mice

All animal procedures were subject to approval by the Institutional Animal Care and Use Committee at Hunan University. We acquired six-week-old female BALB/c mice from Hunan SJA Laboratory Animal Co., Ltd., Changsha, China. In our study, we employed *Klebsiella pneumoniae* as the bacterial strain to induce infection in the mice. The mice were randomly divided into three groups: the PBS group, the green AgNP group, and the chemical AgNP group. We anesthetized all the mice using chloral hydrate and created a circular skin wound with a 10 mm diameter on their backs. Subsequently, we applied a 100 μL suspension of *Klebsiella pneumoniae* (at a concentration of 10^8^ CFU mL^−1^) to infect the circular wounds. After three days of continuous infection, we treated the wounds with 100 μL of PBS, green AgNPs, and chemical AgNPs. Concurrently, we measured the wound diameter using a digital vernier caliper and documented the wound’s progress every two days. Following ten days of treatment, we collected skin tissue samples, diluted, and homogenized them, and plated aliquots on agar. The colonies that grew were then counted using standard plate counting methods for further analysis.

On the 12th day, all mice were euthanized, and their skin tissues and major organs were removed to evaluate their healing efficacy and assess biological safety. For histological analysis, we harvested wound tissues from each group of mice and fixed them in a 4% paraformaldehyde solution. These samples were prepared for subsequent staining and H&E analysis. To assess the in vivo antibacterial effect, infected tissues were collected for coating assays. Blood samples were also drawn from the treated mice after the in vivo wound healing experiments, and these samples were analyzed biochemically.

### 2.14. Statistical Analysis

The data were presented in the form of mean values along with their corresponding standard deviations, and each experiment was conducted a minimum of three times. *T*-test or one-way (ANOVA) test using the GraphPad Prism 8.0.2 software was applied to test the significance between groups where significance levels were denoted as follows: * *p* < 0.05, ** *p* < 0.01, *** *p* < 0.001, and **** *p* < 0.0001 indicating that there is a statistically significant difference between various groups.

## 3. Results

### 3.1. Preparation and Characterization of Silver Nanoparticles

The synthesis strategies of AgNPs by green and chemical step methods are schematically illustrated in Figure 1 and Figure 1A. Firstly, fresh leaves from coriander were selected, washed thoroughly, and dried for extraction. After 24 h stirring, stable and biosynthesized AgNPs were obtained successfully. On the other hand, AgNPs from ascorbic acid and sodium citrate were prepared chemically. The preparation of green AgNPs was more environmentally friendly than the chemical method. The UV-vis absorption spectra showed the corresponding peaks of green AgNPs (approximately 430 nm) and chemical AgNPs (420 nm) (Figure 1B).

The zeta potential of AgNPs extracted from coriander plants and chemically synthesized were −32.4 mV and −23.4 mV, respectively, which directly reflected successful and stable synthesis of AgNPs (Figure 1C). Meanwhile, as measured by dynamic light scattering (DLS), the dimensions of chemical and green AgNPs were ~42 nm and ~167 nm, respectively (Figure 1D). TEM images showed cuboidal shape of green AgNPs with diameter of approximately 100 nm and chemically AgNPs (Figure 1E,F), respectively. The chemical composition of nanomaterials was investigated by X-ray photoelectron spectroscopy (XPS). The binding energy peaks of green AgNPs located at 294.05 eV, 378.2 eV, 406.25 eV, and 541.5 eV were ascribed to C1s, Ag3d5, N1s, and O1s, respectively (Figure 1G).

The FTIR spectra of green AgNPs revealed several characteristic peaks, each corresponding to specific functional groups or chemical bonds present on the nanoparticle surface. An intense and broad absorption peak at 3401 cm^−1^ signifies O-H stretching vibrations, typically linked to hydroxyl groups (alcohols and phenols) or adsorbed water molecules on the nanoparticle surface. Peaks observed at 2922 cm^−1^ and 2823 cm^−1^ correspond to C-H stretching vibrations, suggesting the existence of organic compounds or hydrocarbons on the surface of the AgNPs. The presence of a peak at 1850 cm^−1^ hints at the potential adsorption of carbon-containing species. A peak at 1625 cm^−1^ indicates C=C stretching vibrations, potentially stemming from unsaturated organic molecules. The presence of a peak at 1581 cm^−1^ points to C=O stretching vibrations, suggesting the existence of carbonyl groups, likely originating from organic capping agents. Another noticeable peak at 1440 cm^−1^ suggests the likelihood of CH_2_ bending vibrations, typically associated with aliphatic hydrocarbons. The peak at 1214 cm^−1^ aligns with C-O stretching vibrations, indicating the presence of alcohols, ethers, or esters on the surface of AgNPs. The peak at 1022 cm^−1^ corresponds to C-N stretching vibrations, possibly indicating the presence of amines or amides. Overall, these FTIR findings suggest that the Ag nanoparticles are enveloped or associated with diverse organic compounds and functional groups on their surface (Figure 1H).

In this study, the HPLC method with multiple wavelengths was used to observe any changes that could affect our results. Three different wavelengths (275, 284, and 360 nm) were used to detect AgNPs and other compounds present in the coriander plant extract, AgNPs extracted from coriander, and AgNPs synthesized from ascorbic acid by chemical methods. Chemical AgNPs showed reduced Ag products as peaks were shown around 30 min at 360 nm (Figure 2A). There were some active compounds observed in the coriander plant extract which have been reported in published data (Table 1), but there was no active compound detected in the Ag sample except Ag. At 360 nm, we observed very small peaks around 4–10 min. That matches the same retention time as observed plant extract as well observed in published data (Figure 2B). In green AgNPs samples, the lack of other peaks mean plant compounds reduced the Ag^+^ to Ag^0^ (Figure 2C). There were no residues in the plant synthesized AgNPs, and as such they are more suitable to be used in vivo as well in vitro studies as HPLC is limited in detention. We observed similar results at each wavelength, proofing that our samples met the same criteria.

### 3.2. Assessment of In Vitro Antibacterial Experiment

The in vitro antibacterial activity of chemical and green AgNPs was considered by bacteriostatic ring test. After 24 h of culture, no inhibition ring was observed in the bacteria after treatment with water and hydrogel (Figure 3A). By contrast, green AgNPs, green AgNPs (hydrogel), and chemical AgNPs treatment groups demonstrated apparent inhibition zones 20.04 mm, 19 mm, and 8 mm, respectively, owing to the antibacterial activity (Figure 3B). Green AgNPs indicated substantially larger inhibition zones than chemical AgNPs. These results demonstrated that green AgNPs have a stronger inhibitory effect against Gram-negative bacteria *Klebsiella pneumoniae* compared to chemically synthesized AgNPs.

The minimum inhibitory concentrations (MICs) of different nanomaterials against *Klebsiella pneumoniae* were also assessed. According to the data, the MICs of green and chemical AgNPs were 8 µg/mL and 6 µg/mL, respectively. Green AgNPs showed the lowest concentration (8 µg/mL) after incubation, yielding no visible bacterial growth, whereas chemical AgNPs showed the highest bacterial growth (Figure 3C). Therefore, it is concluded that green AgNPs dramatically improve the antimicrobial activity compared to chemically synthesized AgNPs.

The plate counting method was adopted to test in vitro antibacterial efficiency of samples. *Klebsiella pneumoniae* with different treatments was diluted 10^5^ and 10^3^ times, respectively, before they were spread on the agar plates. Figure 3D denoted bacterial colonies for green AgNPs, chemical AgNPs, and vancomycin treatment groups, compared with the control group, which was stable with an agar diffusion test. These results implied that chemical AgNPs could not completely eradicate the bacteria whereas no bacterial colonies were observed at 50 and 60 µg/mL green AgNPs. Furthermore, OD_600_ explained that green AgNPs have a stronger antibacterial effect than chemical AgNPs (Figure 3E). Figure 3F showed the bacterial survival rate of different nanoparticles. The results showed that ~97.30% *Klebsiella pneumoniae* cells were killed by green AgNPs and ~26.77% *Klebsiella pneumoniae* cells were killed by chemical AgNPs.

The sustainability of *Klebsiella pneumoniae* was considered using fluorescence staining assay. Confocal images of the live/dead staining assay indicated that only strong green fluorescence was observed in bacteria after treatment with PBS, cefixime, and chemical and green AgNPs treatment groups. In comparison, PBS-, cefixime-, and chemical AgNP-treated bacteria instantaneously produced green fluorescence, implying partial bacteria-killing efficacy was attained. Conversely, bacteria treated with green AgNPs showed extremely weak green fluorescence, which reflected strong bacterial killing ability (Figure 4A). The quantitative assay of fluorescence intensity using Image J 1.52v software directly indicated the highest antibacterial efficacy of green AgNPs compared with the other groups (Figure 4D).

SEM images of Figure 4B similarly established negligible antibacterial ability of PBS and chemical AgNPs treatment, which was supported by the intact and smooth bacterial membranes, while the bacterial membranes with green AgNPs treatment were obviously shrinking. 

### 3.3. Antibacterial Mechanism Exploration

The antibacterial mechanism was then investigated by utilizing a DCFH-DA probe to observe ROS generation in *Klebsiella pneumoniae*. Green AgNPs showed the strongest green fluorescence signal, representing the highest level of ROS (Figure 4C,E), whereas chemical AgNPs and cefixime produced weak and less green fluorescence than green AgNPs, indicating the production of a smaller amount of ROS. Furthermore, higher levels of ROS were produced in bacteria treated with green AgNPs due to spontaneous release of Ag^+^.

We also investigated ATP level in bacteria with green and chemical AgNPs treatment. Figure 4F indicated that the ATP level in bacteria cultivated with green AgNPs was significantly lower than that of chemical AgNPs, which was decreased by 87%, indicating the superior antibacterial efficacy of this kind of approach, whereas chemical AgNPs showed a weak and insignificant decrease in ATP.

Our study also observed leakage of protein as well as nucleic acid membrane. The results of nucleic acid (DNA and RNA) and protein leakages level assay (Figure 4G–I) respectively indicated that green AgNPs rigorously destructed bacterial cell membranes more efficiently than chemical AgNPs. Green AgNPs cause bacterial morphological distortion, leaking of nucleic acids (RNA, DNA) and proteins, and a decrease in ATP levels. Through mechanisms involving morphological damage, metabolic disturbance, and respiratory depression, these combined impacts ultimately caused bacterial death.

### 3.4. In Vitro Dispersion of Established Klebsiella Pneumoniae Biofilms

The formation of biofilm involves a three-stage process, as illustrated in Figure 5A. Initially, bacteria adhere to the medium surface, creating a conditional film. Subsequently, bacteria aggregate together. Lastly, bacteria undergo proliferation, giving rise to well-structured three-dimensional configurations featuring channels and voids.

Considering the tremendous antibacterial performance of green and chemical AgNPs, we assessed their effect on the developed biofilms of *Klebsiella pneumoniae*. From Figure 5B, we found that the green AgNPs treatment sample had only a thin layer of residual bacteria persist within the biofilm, and the broad biofilm dissipated, leaving only a few residual bluish spots as an indication of the presence of residual bacteria, whereas the chemical AgNPs treatment sample had dense biofilm clumps. In comparison, green AgNPs (40 and 50 µg/mL) eradicated 68% and 75% biofilm mass, respectively, whereas chemical AgNPs with same concentrations eliminated 47% and 52% biofilm mass, respectively (Figure 5C).

In summary, green AgNPs not only demonstrated strong bactericidal effects on free-floating bacteria but also showed a remarkable ability to eliminate biofilms. This makes them a promising option for promoting the healing of chronic wounds compared to chemically produced AgNPs.

### 3.5. Enzyme-Based Antioxidant and DPPH Assay

This study evaluated the antioxidant activity of as-prepared AgNPs using an enzyme-based approach, employing TMB as a peroxidase substrate. Over time, the UV-vis absorption intensity progressively increased, indicating the oxidation of TMB in the presence of AgNPs. In Figure 6A, UV-vis spectrophotometry results revealed that the control group exhibited no oxidation (depicted by the blue curve), while the green AgNPs (represented by the green curve) exhibited a higher absorption intensity compared to chemically synthesized AgNPs (illustrated by the red curve). The absorption intensity versus time graph for these systems is presented in Figure 6B. The UV-vis absorption intensity continued to rise over time, signifying oxidation in the presence of green AgNPs and H_2_O_2_, whereas chemical AgNPs demonstrated a lower intensity curve compared to green AgNPs, suggesting that green AgNPs exhibited stronger oxidation than chemically synthesized AgNPs when using TMB as a substrate.

In this study, the results showed that green AgNPs have stronger antioxidant activity than chemical AgNPs (Figure 6C). The DPPH assay is included as it is a very simple test system which gives a first indication of the radical scavenging potential of a test compound. The DPPH scavenging capabilities of green synthesized AgNPs reached 88% at a concentration of 25 μg/mL, whereas their chemically synthesized counterparts exhibited only 80% scavenging at the same concentration. The heightened antioxidant performance of green synthesized AgNPs can be attributed to the surface modification facilitated by phytochemicals found in coriander plant leaf extracts. Figure 6C illustrated the DPPH scavenging activity of both green and chemical AgNPs. Notably, the percentage of DPPH scavenging activity increased linearly with rising nanoparticle concentrations, ranging from 5 to 25 μg/mL, for both the green and chemical AgNPs variants. This discrepancy in antioxidant activity underscores the superiority of nanoparticles synthesized with coriander leaf extract due to the presence of a bioactive capping agent on their surface. The synergistic combination of antioxidant and antibacterial characteristics of our developed green AgNPs offer potential applications in both biomedical and environmental fields.

### 3.6. In Vitro Cytotoxicity Study

The hemolysis rates of green AgNPs at different concentrations (2, 4, 6, 8, and 10 µg/mL) were 0.14%, 5.59%, 10.54%, 2.18%, and 11.18%, respectively. In contrast, chemical AgNPs exhibited hemolysis rates of 16.99%, 25.13%, 38.82%, 37.21%, and 42.16% at the same concentrations as green AgNPs. These results indicate that green AgNPs demonstrated superior biocompatibility compared to chemical AgNPs (Figure 7A,B) and it is clearly indicated that green AgNPs did not induce hemolysis, but chemical AgNPs triggered hemolysis. Furthermore, green AgNPs showed no significant effect on their morphology, maintaining a pronounced bi-concave disc shape like that of PBS, while chemical AgNPs displayed some atypical morphological changes (Figure 7C). Additionally, the cytotoxicity assay in Figure 7D showed that the cell viability of green AgNPs after incubation for 24 has an ultra-low cytotoxicity effect compared to chemical AgNPs. However, similar viability was found in Figure 7D between green and chemical AgNP groups. Phytogenic green synthesized AgNPs using coriander leaves were found to be safer than chemically synthesized AgNPs in cytotoxicity assay on mice.

### 3.7. In Vivo Evaluation of Wound Healing

In this paper, we assessed the impact of chemical and green AgNPs on wound healing of mice. Female BALB/c mice with *Klebsiella pneumoniae*-infected wounds on their back were used as the model for this study. Figure 8A explains the descriptive photographs of wound closure with different treatments at different time periods. The representative photographs of wound closure after various treatments at different time points are displayed in Figure 8B. After healing for 12 days, the area of infected wounds in PBS, chemical AgNP, and green AgNP groups decreased by 52.81%, 45.94%, and 26.75% respectively, signifying very limited upgradation of single-mode treatment to the wound healing (Figure 8C), which clearly indicated that green AgNPs have stronger wound healing efficacy than chemical AgNPs. The traces of wound healing for 12 days for each group were also drawn (Figure 8D). No significant change of body weight was observed for green AgNPs (Figure 8E), reflecting the biosafety of green AgNPs for mice. The in vivo antibacterial effect was also evaluated by comparing the bacterial CFU in the infected area. The in vivo anti-*Klebsiella pneumoniae* effect assay indicated that green AgNPs exhibited the best antibacterial effect in a *Klebsiella pneumoniae*-infected wound compared to chemical AgNPs and PBS (Figure 8F,G).

Upon conducting an inflammation analysis using hematoxylin and eosin (H&E) staining, this study revealed a significant presence of inflammatory cells on the wounds treated with chemical AgNPs. In contrast, when the wounds were treated with green AgNPs, there was a noteworthy reduction in the number of inflammatory cells, and the wound closures in the green AgNP group closely resembled the structure of normal tissue. In the PBS group, the epidermis appeared thin and was stained in a blue-violet shade, with a replacement of the original pink connective tissue underneath and extensive infiltration of lymphocytes. In the chemical AgNP group, the epidermis remained thin, exhibiting a violet-blue hue, with lower lymphocyte infiltration and the presence of pink connective tissue around the wound area. However, the wound healing was incomplete. In contrast, the green AgNP group displayed a thin, purplish-blue epidermis, with the damaged surface layer of the wound area, but the tissue had fully healed (Figure 9A).

Additionally, Masson’s trichrome staining revealed that the regenerated collagen fibers (depicted in blue) in the green AgNP-treated group were continuous and more prominent compared to other treated groups. This finding indicated a superior recovery in wound healing. After 12 days, skin sections from animals in the PBS-, chemical AgNP-, and green AgNP-treated groups showed an epidermis with a coagulated mass of blood, resulting in the rupture of the dermis layer. Dark blue collagen fibers were observable at this site, as illustrated in Figure 9A.

Furthermore, within the PBS group, the epidermal layer exhibited a red staining pattern, with fibroblasts and inflammatory cells predominantly located beneath the affected areas, as visualized through Masson’s trichrome staining (Figure 9B). Similar red staining of the epidermis was observed in the chemical AgNP group, where some fibroblasts, inflammatory cells, and traces of collagen fibers were still detectable beneath the lesion. Notably, insufficient collagen fiber deposition was observed in the neighboring tissues, suggesting incomplete healing. Conversely, in the green AgNP group, the epidermis displayed a red staining pattern, and a substantial amount of collagen fibers had deposited below the lesion. The surrounding tissues exhibited dark blue collagen fibers and dark red muscle fibers, indicating that the lesion was nearing complete healing.

As previously discussed, we delved into the rationale for administering these substances to enhance the wound healing process. Due to the upregulation of CD31, immune stained skin slices from green AgNPs exhibited a notably strong CD31 expression in the endothelial cells of the newly formed capillaries. In contrast, chemical AgNPs displayed a relatively less pronounced CD31 expression compared to green AgNPs. The treatment group receiving green AgNPs significantly outperformed both the PBS and chemical AgNP groups in terms of the positive CD31 and VEGF signals, as quantified and depicted in Figure 9C and 9D, respectively.

Remarkably elevated levels of VEGF play a crucial role in stimulating the formation of new blood vessels in damaged tissue. This heightened VEGF-enhanced metabolic activity, cell proliferation, and the re-establishment of epithelial layers all contribute to effective wound healing through the assistance of newly established vasculature. Immunohistochemistry (IHC) staining of infected wounds treated with green AgNPs revealed the highest VEGF expression among all the groups (Figure 9D).

### 3.8. In Vivo Biological Safety Study

We have also investigated the effect of green and chemical AgNPs on the microscopic structure of liver, heart, kidney, spleen, and lung tissues in female mice using H&E staining. Strong antibacterial activity against microorganisms and acceptable systemic safety are essential for effective clinical translation of antibacterial materials. When the wound healing trials were finished, a biochemical study of the mice after various treatments was carried out to determine the in vivo toxicity of AgNPs. There were no significant differences observed among all groups in terms of total white blood cell (WBC), total red blood cells (RBC), hemoglobin (HGB), and blood platelet (PLT) counts. However, green AgNPs demonstrated superior and more effective outcomes in biochemical assays compared to chemically synthesized AgNPs, as depicted in Figure 10A–D. Furthermore, essential hepatic and kidney indicators, including ALT, AST, UREA, and CREA, showed no abnormalities, as illustrated in Figure 10E–H.

Moreover, histopathological examinations using H&E staining of major organs such as the heart, liver, spleen, kidneys, and lungs confirmed the preservation of normal tissue structures without any evident signs of inflammation or organ damage following treatment with green AgNPs, as seen in Figure 10I. All these findings strongly support the superior biosafety of our developed green AgNPs for in vivo infected wound healing therapy in comparison to chemical AgNPs. These results underscore the potential of green AgNPs as an effective and safe platform for treating *Klebsiella pneumoniae* infections, with promising prospects for future clinical applications.

## 4. Discussion

AgNPs were effectively produced through an environmentally friendly method employing coriander leaf extract as both a reducing and capping agent. Coriander AgNPs contain various bioactive compounds, including flavonoids and polyphenols, which have synergistic effects with AgNPs in promoting wound healing and reducing inflammation. AgNPs from coriander exhibit improved biocompatibility, reducing the risk of adverse reactions when applied to wounds compared to chemically synthesized AgNPs. Biosynthesized AgNPs were characterized by different methods. UV-vis spectrophotometry showed that the AgNPs were stable and showed peaks of chemical and green AgNPs (Figure 1B). The results verified that the green and chemical AgNPs were indeed crystalline in nature, and morphological examinations revealed their cuboidal shape with sizes ranging from around 100 nm (Figure 1E,F). The antibacterial efficacy of biosynthesized AgNPs against Gram-negative bacteria is often enhanced due to the characteristics of the peptidoglycan layer present in their cell walls [29]. Recent research indicated that incorporating supporting components in the form of nanocomposites to reduce agglomeration can effectively augment the antibacterial activity of AgNPs [33]. The bactericidal effects of nanomaterials play an essential role in cytoplasmic leakage, leading to bacterial cell death [34]. Figure 3 demonstrated that green AgNPs have a stronger inhibitory effect against Gram-negative bacteria *Klebsiella pneumoniae*.

Previous studies reported that metal nanomaterials could influence microbial antioxidant defense systems and lead to cell damage by generating significant levels of reactive oxygen species (ROS) [35,36]. A DCFH-DA probe was used to investigate ROS in this study. Figure 4C,E clearly showed that green AgNPs have higher ROS generation. ATP, the most important energy molecule, contributes to many physiological and pathological activities of bacteria. Mostly, ATP will reduce under conditions of necrosis or apoptosis [37]. We also investigated ATP level in green and chemical AgNPs. Green AgNPs showed a decline in ATP level which indicated that green AgNPs have superior antibacterial activity (Figure 4F). The devastation of the bacterial membrane and cytoplasm leakage were also explored by quantitative analysis of protein leakage, which is interpreted as a descriptive indicator of cell-content leakage [38]. Our study also observed protein as well as nucleic acid membrane leakage. The overall results showed that green AgNPs destructed robustly bacterial cell membrane (Figure 4G–I).

The formation of biofilms, which can hinder the penetration of drugs and escape the body’s natural immune defenses, is closely tied to the persistence of bacterial infections [39,40]. Therefore, eliminating biofilm of bacteria is more challenging than eliminating planktonic bacteria [40,41]. These biofilms are strengthened by the production of sticky extracellular polymers (EPS), which bolster the microorganism’s ability to withstand attacks from the host’s immune system and external antibiotics, ultimately offering protection to the bacteria [42]. Therefore, we also investigated biofilm removal by our green and chemical methods. The results showed that green AgNPs have a more remarkable capability to eliminate biofilm than chemical AgNPs (Figure 5).

Enzyme-based antioxidant and DPPH assay were also investigated to find out the antioxidant properties of both green and chemical AgNPs. The catalytic process can be elucidated through the oxidation of TMB by hydroxyl radicals (·OH). In an acidic environment, AgNPs break down hydrogen peroxide (H_2_O_2_) to generate hydroxyl radicals (·OH) and transform the initially colorless TMB into a blue-colored compound known as oxTMB [43,44]. The catalytic mechanism proposed by Tran et al. [45] assumes that H_2_O_2_ dissolves AgNPs, releasing Ag^+^ that will then oxidize the colorless TMB solution to blue-colored oxTMB. DPPH is a stable free radical with a distinctive purple hue and has a prominent absorption peak at 517 nm. When an antioxidant is introduced, it couples with the free radical within DPPH, leading to a decrease in both absorbance and color intensity. In the presence of AgNPs, electron or proton donation occurs to reduce the DPPH radicals [46]. Our results showed that green AgNPs have stronger antioxidant activity than chemical AgNPs (Figure 6C). This discrepancy in antioxidant activity underscores the superiority of nanoparticles synthesized with coriander leaf extract due to the presence of a bioactive capping agent on their surface. The combination of antioxidant compounds from coriander extract with the antibacterial properties of AgNPs have exhibit synergistic effects.

The biocompatibility of a nanodrug is the most important precondition of clinical translation [47]. Furthermore, the in vitro erythrocyte aggregation assay demonstrated that these green synthesized AgNPs did not induce hemolysis (Figure 7A,B). Figure 7D suggested better biocompatibility of green AgNPs than chemically synthesized AgNPs.

AgNPs have several advantageous characteristics for the management of wounds, such as antibacterial, antifungal, and anti-inflammatory effects [48]. A recent study has reported on the multiplication of stem cells for the purpose of promoting healing and reducing the effects of ageing [49]. The chemical reduction approach is commonly utilized for the manufacture of AgNPs. Nevertheless, there are certain drawbacks associated with this approach, including the utilization of potentially hazardous compounds like hydrazine and the tendency for agglomerates to form during storage [50,51]. Due to the disadvantages associated with chemical approaches, there is a growing trend towards the use of biological methods in the creation of nanoparticles.

We also investigated the wound healing acceleration of green and chemical AgNPs. Female BALB/c mice with *Klebsiella pneumoniae*-infected wounds were used in this study. The results, seen in Figure 8, showed significant inflammatory cells on wounds treated with control and chemical AgNPs, whereas wounds treated with green AgNPs showed notable wound closure. The process of wound healing encompasses multiple mechanisms, including epithelialization, contraction, and deposition of connective tissue/matrix. The extracellular matrix is composed of several components, including collagen, elastin, fibronectin, laminin, hyaluronic acid, and proteoglycans. These structures confer mechanical stability and facilitate cellular processes such as expansion, contraction, cell migration, and essential biochemical reactions [52]. In this model, the green AgNPs group had a statistically significant increase in wound contraction and a decrease in the duration of epithelialization, as illustrated in Figure 8.

The inflammatory process plays a pivotal role in facilitating tissue damage recovery and regeneration. The transition from inflammation to the healing of wounds relies on the buildup and activation of inflammatory cells. While a moderate level of inflammation is advantageous for the healing of wounds, an excessive presence of inflammatory cells can impede progress [53]. We detected the levels of CD31 and VEGF, two cytokines associated with angiogenesis and endothelial cell growth, to illustrate the recovery status of wounds. PECAM-1, also referred to as CD31, is a 130 kDa type I transmembrane glycoprotein adhesion molecule belonging to the immunoglobulin superfamily [54]. VEGF, a crucial element that stimulates neo-angiogenesis, plays a vital role in tissue regeneration by providing nourishment and oxygen, making it indispensable for the process of wound healing [55]. The results shown in Figure 9C, D indicate that green AgNPs expedited and enhanced the healing of wounds in mice by increasing the expression of VEGF and CD31, ensuring a rapid and secure healing process.

Recently, studies have shown that exposure to AgNPs can lead to its availability in the blood, and subsequently their distribution throughout various body organs such as the kidneys, liver, spleen, brain, and lungs [56]. They can also activate inflammatory responses by inducing the production of cytokines and chemokines, which can further exacerbate liver damage [57]. Therefore, we also investigated the effect of green and chemical AgNPs on the microscopic structure of liver, heart, kidney, spleen, and lung tissues in female mice using H&E staining. Figure 10 A–D shows that green AgNPs have insignificant differences among all groups in WBC, RBC, HGB, and PLT counts. Meanwhile, no abnormalities were found in green AgNPs-treated groups for key functional hepatic and kidney indicators, such as ALT, AST, UREA, and CREA (Figure 10E–H). In addition, H&E staining images of the major organs, including heart, liver, spleen, kidney, and lung, confirmed normal tissue structures without any visible organ damage (Figure 10I). All these results strongly demonstrate that green AgNPs have superior biosafety for in vivo infected wounds.

In addition, we performed a comparative examination of the UV peak, zone of inhibitions, zeta potential, and size of our approach in relation to previous research (Table 2).

The idea of evaluating the antibacterial characteristics and wound healing capacity of both plant-derived and chemically manufactured silver nanoparticles is highly creative. This research integrates both natural and synthetic methodologies to investigate the advantages of silver nanoparticles, an area of biomedical research that has attracted considerable attention.

Plant extracts are utilized to generate phytogenic silver nanoparticles. The utilization of this green synthesis strategy is ecologically sustainable and may exhibit lower levels of toxicity in comparison to traditional chemical procedures. Plants serve as reducing agents to convert silver ions into nanoparticles, while also capping and stabilizing these nanoparticles, often enhancing their biological activity. Chemogenic silver nanoparticles, however, are produced using chemical synthesis techniques. These approaches frequently provide enhanced manipulation of the dimensions and morphology of the nanoparticles, which is essential for optimizing their antibacterial efficacy. Nevertheless, these techniques may entail the use of toxic substances and might generate perilous secondary substances.

Evaluating both categories of silver nanoparticles for wound healing is innovative as it amalgamates the benefits of both synthesis techniques. Through the comparison of their efficacy and safety in wound healing applications, researchers have the potential to produce more potent treatments for wounds, particularly those that are susceptible to infection.

This type of research is at the vanguard of nanomedicine and has the potential to bring about substantial progress in wound treatment and infection management. It could result in accelerated wound healing, decreased reliance on antibiotics, and improved overall results for patients.

## 5. Conclusions

AgNPs from coriander exhibit improved biocompatibility, reducing the risk of adverse reactions when applied to wounds compared to chemically synthesized AgNPs. The results verified that the AgNPs were indeed crystalline in nature, and morphological examinations revealed their cuboidal shape with sizes ranging from around 100 nm. The FTIR analysis confirmed the presence of functional groups that likely play a role in the bio-reduction and stabilization of Ag^+^ ions during the green synthesis of AgNPs. Furthermore, the in vitro erythrocyte aggregation assay demonstrated that these green synthesized AgNPs did not induce hemolysis.

Investigations into their antibacterial properties unveiled the remarkable ability of green AgNPs to effectively combat *Klebsiella pneumoniae* bacteria by compromising cell integrity, inducing ROS production, reducing ATP levels, and disrupting bacterial metabolism. Green AgNPs from coriander can be used in biomedical applications to combat bacterial infections and in food packaging to extend the shelf life of products. Green AgNPs can also be used in dental materials, surgical implants, bone grafting, wound dressings, and ointments. In conclusion, green AgNPs are preferred over chemical AgNPs because of their more environmentally friendly synthesis, reduced toxicity, affordability, improved biological activity, sustainability, improved control over the properties of the nanoparticles, decreased chance of contamination, and suitability for wide-ranging applications.

## Data Availability

Data are contained within the article.

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
