# Peer review of "Assessing Phytogenic and Chemogenic Silver Nanoparticles for Antibacterial Activity and Expedited Wound Recuperation"

_nanomaterials, 2024, doi:10.3390/nano14030237_

Round 1

Reviewer 1 Report

Comments and Suggestions for Authors

The manuscript "Assessing Phytogenic and Chemogenic Silver Nanoparticles for Antibacterial Activity and Expedited Wound Recuperation" a hard work in the subject of the proposed research.

The presented results are interesting and useful.

However,

There are many works in this field and therefore a comparative table of the results of other researches would be useful. On this occasion, the references section can also be expanded.

Many biological extracts have been used for the preparation of silver nanoparticles. What is the composition of the extract used? At this moment it is unclear (which of the components determines the formation of nanoparticles).

How do the authors explain the very good anti-bacterial results with 100nm "nanoparticles"?

The novelty of the work must be redone.

Reviewer 2 Report

Comments and Suggestions for Authors

The work focuses on the AgNP synthesis using Coriandrum leaves with antibacterial and anti-inflammatory properties. The authors used different characterization techniques using a series of in vitro and in vivo assays. Although the work is a well-presented study, the novelty of that seems insufficient. Other studies, such as the work by Roua Alsubki et al., "Green synthesis, characterization, enhanced functionality and biological evaluation of silver nanoparticles based on Coriander sativum" (Saudi Journal of Biological Science, 28 (4) 2021, 2102-2108, https://doi.org/10.1016/j.sjbs.2020.12.055) and Asma Ashraf et al. "Synthesis, characterization, and antibacterial potential of silver nanoparticles synthesized from Coriandrum sativum L." (Journal of Infection and Public Health, 12 (2), 2019, 275-281, https://doi.org/10.1016/j.jiph.2018.11.002), have previously explored the synthesis of AgNPs using coriandrum leaves and their antimicrobial activity. The authors are encouraged to emphasize the unique aspects introduced in their work compared to previous studies.

Reviewer 3 Report

Comments and Suggestions for Authors

Dear authors,

The manuscript entitled „Assessing Phytogenic and Chemogenic Silver Nanoparticles for Antibacterial Activity and Expedited Wound Recuperation” Ahmad et al. describes the synthesizes AgNPs from coriander leaves and evaluated their antibacterial, anti-inflammatory, antioxidant, and wound-healing acceleration properties in comparison to chemically synthesized AgNPs.

After reading the manuscript, I did not notice any significant errors, nor spelling and stylistic errors, English language and style are also fine.

Some clarifications are required:

 Line 149. In methods section 2.5 In Vitro Antibacterial Experiment needs to explain what OD values have been used.

 Line 83 Coriandrum sativum please italicize.

 Line 28, 586 in vivo please italicize.

Line 326, 343, 429, 710 in vitro please italicize.

Line 318, 508 in vivo and in vitro please italicize.

Line 169, 170, 560, 583, 664 pneumoniae please write from lower case.

Round 2

Reviewer 1 Report

Comments and Suggestions for Authors

The revised manuscript "Assessing Phytogenic and Chemogenic Silver Nanoparticles for Antibacterial Activity and Expedited Wound Recuperation" reponds at corrections and suggestions requested.

The work can be considered for publish!

Reviewer 2 Report

Comments and Suggestions for Authors

The manuscript is suitable for publication in its present from.